# Stratospheric ozone measurements at Arosa (Switzerland): History and scientific relevance

Johannes Staehelin [1], Pierre Viatte [2], Rene Stübi [2], Fiona Tummon [1], Thomas Peter [1]

[1]    Institute for Atmospheric and Climate Science, ETHZ, Zürich
[2]    Federal Office of Meteorology and Climatology MeteoSwiss, Payerne

*Correspondence to*: Johannes Staehelin (johannes.staehelin@env.ethz.ch)

**Abstract.** In 1926 stratospheric ozone measurements were started at the Light Climatic Observatory (LKO) of Arosa (Switzerland), marking the beginning of the world's longest series of total (or column) ozone measurements. They were driven by the recognition atmospheric ozone is important for human health, as well as by scientific curiosity about what was, at the time, an ill characterized atmospheric trace gas. From around the mid-1950s to the beginning of the 1970s studies of high atmosphere circulation patterns that could improve weather forecasting was justification for studying stratospheric ozone. In the mid-1970s, a paradigm shift occurred when it became clear that the damaging effects of anthropogenic Ozone Depleting Substances (ODSs), such as long-lived chlorofluorocarbons, needed to be documented. This justified continuing the ground-based measurements of stratospheric ozone. Levels of ODSs peaked around the mid-1990s as a result of a global environmental policy to protect the ozone layer, implemented through the 1987 Montreal Protocol and its subsequent amendments and adjustments. Consequently, chemical destruction of stratospheric ozone started to slow around the mid-1990s. To some extent, this raises the question as to whether continued ozone observation are indeed necessary. In the last decade there has been a tendency to reduce the costs associated with making ozone measurements globally including at Arosa. However, the large natural variability in ozone on diurnal, seasonal, and interannual scales complicates the ability to demonstrate the success of the Montreal Protocol. And, chemistry-climate models predict a "super-recovery" of the ozone layer at mid-latitudes in the second half of this century, i.e. an increase of ozone concentrations beyond pre-1970 levels, as a consequence of ongoing climate change. These factors, and identifying potentially unexpected stratospheric responses to climate change, support the continued need to document stratospheric ozone changes. This is particularly valuable at the Arosa site, due to the unique length of the observational record. This paper presents the evolution of the ozone layer,  the history of international ozone research, and discusses the justification for the measurements in the past, present and into future.

## 1.    Introduction

The world's longest time series of total (or column) ozone observations is from Arosa in the Swiss Alps, made at the "Light Climatic Observatory" (Lichtklimatisches Observatorium, LKO). This long total ozone dataset is extremely valuable for long-term trend analyses of stratospheric ozone. In addition, other important ozone measurements, such as Umkehr and surface ozone measurements were also made at Arosa. Since the 1970s, when anthropogenic stratospheric ozone depletion became a subject of public concern, the measurements at LKO grew in importance (Staehelin et al., 2016). A comprehensive report on the history of the LKO is presently in preparation (Staehelin and Viatte, in prep.). Here we focus on the societal justification for these measurements over the long

history of the LKO, particularly highlighting the link to the development of international stratospheric ozone research. This paper is based on the extensive correspondence by F. W. Paul Götz - ozone pioneer and founder of the LKO - which is stored in the LKO archives located at MeteoSwiss in Payerne, Switzerland, on the annual reports of the "Kur- und Verkehrsverein Arosa" (KVV Arosa, see below), and on other research. Following Staehelin and Viatte (in prep.) we divide the history of LKO into five distinct periods (see Sections 2-6 below). Section 7 looks at the potential pathways into the future of measurements at the LKO. Finally, a summary and conclusions is presented in Section 8.

## 2. Period 1921-1953: Friedrich Wilhelm Paul Götz

### 2.1. Therapy for tuberculosis prior to the availability of antibiotics

The first ozone measurements at Arosa were a part of medical research focused on the treatment of pulmonary tuberculosis (TB). Before modern antibiotics became available (a few years after World War II), TB was considered as a serious illness with high mortality rates. The best available therapy for treating TB at the time was believed to be the "rest cure therapy" (as proposed, e.g. by Karl Turban, one of the leading medical doctors in Davos at the time, see e.g. Virchow, 2004). At the end of the 19th century and the beginning of the 20th century many sanatoria and hotels were constructed in Alpine villages such as Davos and Arosa. During "rest cure therapy", which was more fully developed in the first decades of the 20th century, the patients stayed outside on balconies during the day under strict hygienic conditions, usually for several months at a time. Recovery mainly occurred simply by resting. From a modern medical perspective, such rest under strict hygienic control (in order to prevent reinfection) in special lung clinics was probably indeed the most helpful type of therapy before treatment by antibiotics became possible.

The medical doctors of Davos and Arosa were convinced that the high altitude climate was an important factor for optimal recovery from TB. To study this further, the potentially relevant environmental factors needed to be investigated. Already in 1905, Turban proposed opening an institute aimed to study the scientific effectiveness of the "rest cure therapy" of pulmonary TB (SFI, 1997). However, because of a lack of consensus among medical doctors, this institute was founded only 17 years later in 1922. On 26 March 1922, the municipality of Davos ("Landsgemeinde") decided to create a foundation for an institute for high mountain physiology and tuberculosis research ("Institut für Hochgebirgsphysiologie und Tuberkuloseforschung", today the "Schweizerisches Forschungsinstitut für Hochgebirgsklima und Medizin, SFI" in Davos). The resources for operating the institute mainly originated from a small fee that was paid by all guests of staying in the town, who needed to register when staying in Davos (a form of "tourist tax").

At this point, Carl Dorno played an important role. He was a rich industrialist from Königsberg (Germany), who came to Davos because his daughter suffered from pulmonary TB. She unfortunately passed away a few years after arriving in Davos, but Dorno remained and founded an institute to study the environmental factors important for treating TB using his own funds in 1907 (SFI, 1997). During the first World War and in the subsequent period of inflation, Dorno lost most of his financial resources. On 18 February 1923, the municipality of Davos decided to support the Observatory Dorno, the nucleus of the renoned Physical Meteorological Observatory Davos

(PMOD), which since 1971 also serves as the World Radiation Center (WRC) of the World Meteorological
Organization (WMO), a center for international calibration of meteorological radiation standards within the global
network. When Dorno retired as director in 1926, the institute was integrated as an independent department into
the Swiss research institute for high mountain physiology and tuberculosis research in Davos and was financed by
the Davos community, similar to the other institutes. Despite numerous studies, however, it was never shown that
the Alpine climate was a superior environment for recovery from pulmonary TB (Schürer, 2017).

**2.2.    F.W.P. Götz and the foundation of the LKO (LKS)**

Friedrich Wilhelm Paul Götz grew up in Southern Germany (Göppingen, close to Stuttgart) and went to Davos for
the first time prior to the beginning of the First World War to recover from pulmonary TB, when he was working
on his PhD thesis in astronomy (see Fig. 1). He stayed twice in the "Deutsche Heilstätte" sanatorium (1914-1915)
after which he was released as "fit for work". For the following years (1916-1919) he intermittently taught at the
"Fridericianum" German school in Davos and later worked with Dorno (probably for some months) during the
1919-1920 period. See Staehelin and Viatte (in prep.) for more details.

**Friedrich Wilhelm Paul Götz**

| | |
|---|---|
| 1891 | Born on 20 May in Heilbronn (Germany) |
| 1891-1910 | Childhood in Göppingen (near Stuttgart, Germany) |
| 1910 | Start of Studies in mathematics, physics and astronomy in Heilbronn (Germany) |
| 1914-1915 | Davos: recovery from tuberculosis at «Deutsche Heilstätte» |
| 1916-1919 | Intermittently high school teacher at the «Fridericianum» (German School) in Davos, Switzerland |
| 1919 | Dissertation, University of Heidelberg (Germany), thesis on the photometry of the moon surface |
| 1919-1920 | Part-time coworker of Dorno in Davos |
| 1921 | Founding of Lightclimatic Observatory (LKO) at Arosa |
| 1931 | Habilitation and lecturer at the University of Zürich, Switzerland |
| 1932 | Marries Margarete Karoline Beverstorff (27. Dec.) |
| 1940 | Promotion to «Titular-Professor» at University of Zürich, responsible for teaching courses in meteorology |
| 1950-1954 | Illness (including arteriosclerosis) |
| 1954 | Died on 29 Aug. in Chur (Switzerland) |

**Figure 1.** Biography of F.W. Paul Götz, founder of the Light Climatic Observatory in Arosa.

It appears that Götz was the main driver behind the initiative to make atmospheric measurements at Arosa. He
likely first contacted the Arosa medical doctors and together they subsequently made a request to the managing
committee of the KVV Arosa in March 1921 to initiate climate studies relevant for health. The KVV Arosa (Kur-
und Verkehrsverein Arosa) was an organization that had a fairly large budget. It was supported mainly through
the "tourist" tax,. a fee paid by foreigners/guests staying in Arosa, which was also used to cover the costs of various
other activities that nowadays are subject of communal responsibility. Götz's request was supported by the General
Assembly of the KVV Arosa on 20 August 1921, and Götz was asked to found the "Light Climatic Station" (LKS),
which later became known as the "Light Climatic Observatory (LKO)". The objectives of the LKS were to
complement the meteorological observations made at Arosa since 1884 by the Swiss national weather service (now
"MeteoSwiss") by measurements which were thought to be relevant for studying the recovery from pulmonary
TB. Thus, in 1921 Arosa was the first municipality to finance an institute with the task of studying environmental
factors favorable to curing (pulmonary) TB. The support Götz obtained from the KVV Arosa was rather modest
and he later secured additional regular funding from, the Chur-Arosa railway company, the Arosa municipality
and the canton of Grisons (for more detail see Staehelin and Viatte, in prep.). The LKS measurements were made
on the roof of the Inner-Arosa Sanatorium, where nowadays the "Grand Hotel Tschuggen" is located (see Fig. 2).

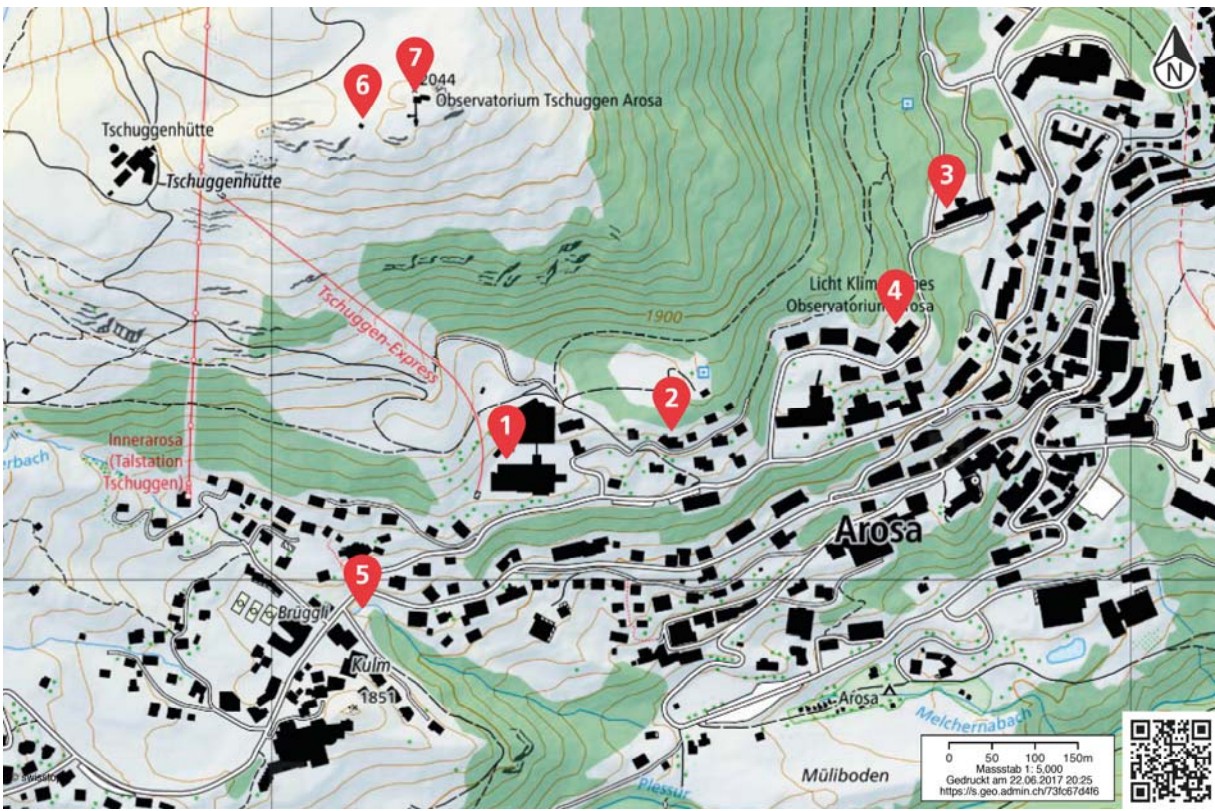


**Figure 2.** Map of important locations relevant to the Arosa Light Climatic Observatory (LKO). LKO measurement
sites: (1) Sanatorium Inner-Arosa; (2) Villa Firnelicht; (3) Florentinum; (4) Haus zum Steinbruch. Other sites: (5)
Götzbrunnen (fountain in honor of Götz); (6) hut where Götz made his nighttime measurements in Tschuggen; 7)
astrophysical observatory at Tschuggen. With permission of swisstopo (Swiss digital maps, geo.admin.ch).


For the first few years Götz was able to borrow an instrument from Dorno (who was based in Davos, see Section
2.1) to measure "biologically active ultraviolet (UV) radiation". This instrument had been adapted and used by
Dorno and consisted of a photoelectric cell with a cadmium (Cd) cathode (Levy, 1932). Götz published several
papers using measurements covering the period November 1921-May 1923 (Götz 1925, 1926a and b). He found
the first indication of seasonal variability of stratospheric ozone in the northern mid-latitudes, with a minimum in
autumn and maximum in spring. This turned out to be a very important result later contributing to develop a better
understanding of stratospheric circulation patterns. This seasonal cycle represents one pillar on which the modern
understanding of the Brewer-Dobson circulation rests. In fact, Götz published this result earlier than the well-
known publication of Dobson and Harrison (1926). Dorno did not agree with Götz's Cd-cell results, and this led
to an open dispute published in the literature (Dorno, 1927). It seems likely that there were also some personal
difficulties between Dorno, who was 26 years older, and Götz, which surfaced with time. It also appears there
were issues between the physicians from Davos and Arosa, with the latter suggesting that the scientific studies
made in Arosa should be coordinated with those from Davos. They also asked that the institute for high mountain
physiology and tuberculosis research in Davos (Institut für Hochgebirgsphysiologie und Tuberkuloseforschung in
Davos) be renamed to include Arosa. These efforts failed probably since members of the Davos community wanted
a larger financial contribution from Arosa for the institute (based on the principle of equal duties, equal rights
("gleiche Rechte, gleiche Pflichten")). The KVV Arosa was, however, not willing to pay the requested amount.

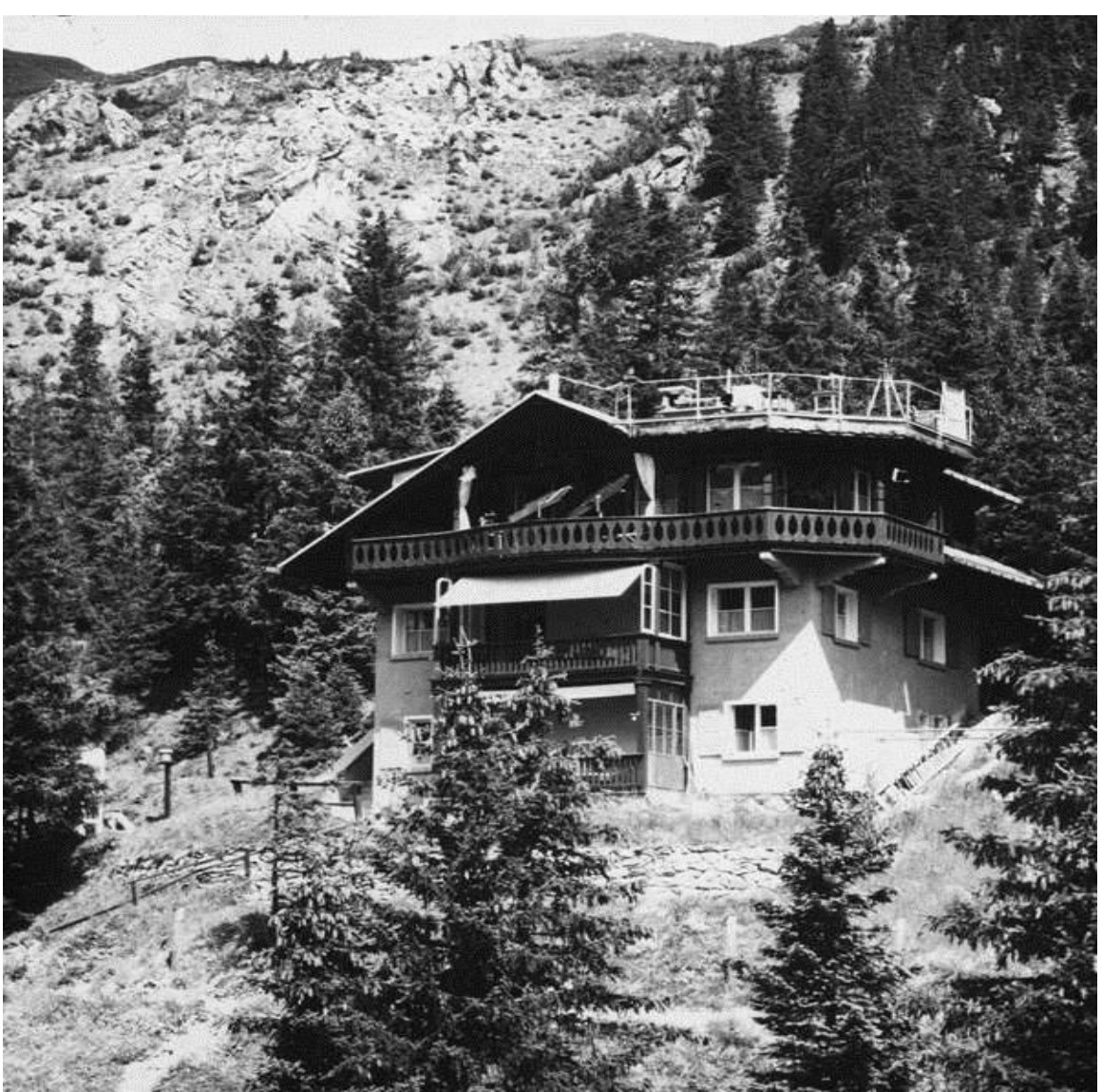

**Figure 3.** "Villa Firnelicht", Götz's house in which the LKO, Götz's observatory was hosted (see text).


## 2.3.    LKO under Götz

1926 was an important year for Götz. After the sobering debate regarding cooperation between the Arosa and Davos medical doctors (for more details see Staehelin and Viatte, in prep.) Götz moved into the "Villa Firnelicht" (see Fig. 3), which is very close to the Inner-Arosa Sanatorium, where measurements had been previously performed (see Fig. 2). Evidence suggests that Götz used family resources to build the large house, probably the inheritance from his father, Paul Götz, who owned an ironmongery ("Eisenwarenhandlung") in Göppingen (Trenkel, 1954) and died in 1926. "Villa Firnelicht" offered space for atmospheric observations on the roof and a balcony. It hosted three apartments and was therefore too large for just Götz and his wife. When Götz moved into "Villa Firnelicht" the institute was renamed the "Light Climatic Observatory" (Lichtklimatisches Obervatorium (LKO)). Götz invited colleagues to come to the LKO for sabbatical-type collaborations and to make atmospheric observations.

**Figure 4.** Sun photometers operated at Arosa from 1921-present (for more details see Staehelin and Viatte in prep.).


Hartley (1881) was the first to postulate that atmospheric ozone is responsible for absorbing solar light in the UV-
B spectrum. Because the amount of biologically active UV-radiation is determined by stratospheric ozone levels,
Götz devoted a large part of his time to stratospheric ozone research (see Staehelin and Viatte, in prep.). He realized
that studying stratospheric ozone required suitable instrumentation and using resources from the KVV Arosa he
mandated the Schmidt-Haensch company based in Berlin (Germany) to construct a Buisson-Fabry type of a sun
spectrophotometer, with a design supervised by him. The instrument was delivered and used by Götz in his
expedition to Spitzbergen (see below), but it is unknown to us why it was subsequently only rarely used. In 1926
Götz started a very fruitful collaboration with Gordon Dobson, a British physicist and meteorologist at the
University of Oxford, who had just developed his first spectrophotometer (Walshaw, 1989). Götz began
continuous total ozone measurements at Arosa using an instrument called a Fery spectrograph, which was
developed by Dobson (Staehelin et al., 1998a). Later, Götz used improved sun spectrophotometers also constructed
by Dobson (abbreviated as Dx, where x is the fabrication number; see Fig. 4). Dobson was very interested in the
favorable climate and good weather and working conditions at the LKO. Thus, he arranged that the instruments
were formally made available to the LKO through the International Association of Meteorology and Atmospheric
Sciences (IAMAS, an association of the International Union of Geodesy and Geophysics (IUGG)). This allowed
Götz to make total ozone observations at Arosa for many years, since it would have been very difficult for him to
buy such spectrophotometers. After 1948 these instruments were formally borrowed through the International
Ozone Commission (IO3C) of the IAMAS. The sun photometers constructed by Dobson measure the intensity of
solar radiation at wavelength pairs in the range of 300-340 nm at the Earth's surface. Three different types of
instruments were constructed by Dobson (Dobson, 1968) which are shortly characterized in Fig. 5. In order to
minimize the falsifying effects of atmospheric aerosols on total ozone measurements the two wavelengths pairs
method was introduced during the International Geophysical Year (1958).
Götz became one of the leading ozone researchers. In the second half of the 1920s and the first half of the 1930s
a key research question was how ozone is distributed in the vertical. Surface measurements e,g, from Arosa
indicated low tropospheric ozone concentrations and rather unprecise measurements suggested ozone maxima in
the mid-latitudes (in partial pressure) at altitudes of around 40-50 km (see Dobson, 1968). The Umkehr method
developed by Götz et al., 1934 (see Fig. 5), however, showed maximum concentrations rather at 20-22 km. This
was considered a scientific breakthrough providing the first reliable information about the vertical ozone profile.
This method is based on the "Umkehr effect", which Götz discovered during his expedition to Spitzbergen in 1929
(Götz, 1931). The first series of Umkehr measurements (besides a limited number of observations made in Oxford
in 1931) was performed together with Dobson and his coworker Meetham on the roof of the "Villa Firnelicht" in
1932/33 (Götz et al., 1934).
Götz was active in the international research community, as a member of the International Radiation Commission
from 1932-1936 (Int. Rad. Com., 2008) and as a member of the International Ozone commission (IO3C) created
in 1948, when it was formally established at the Seventh IUGG Assembly, until 1954 (see Bojkov, 2012). Götz's
research interests were broad, concerning many aspects of weather and climate, and led him to publish two books
on focusing to the statistical analysis of radiation measurement and meteorological observations made at Arosa
(Götz, 1926b; 1954).

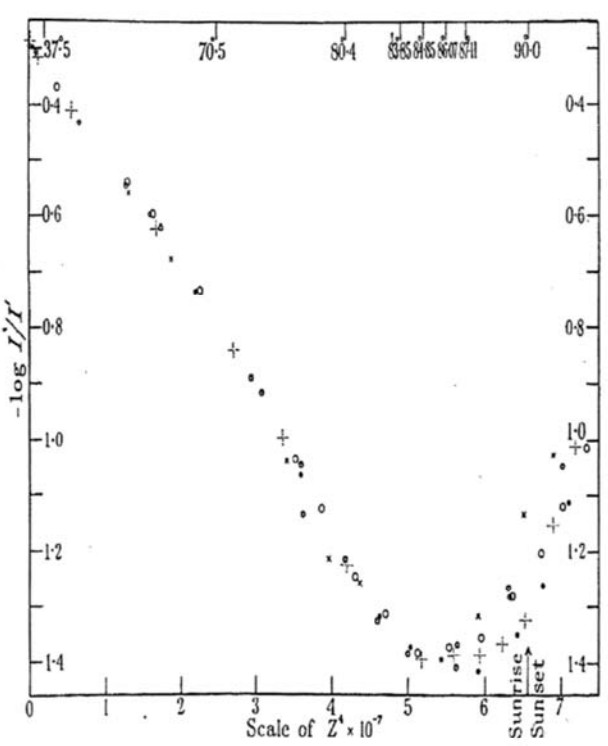

**Ozone profile** by the **Umkehr method**: Zenith sky measurements (wavelengths pair C) as function of time including sunrise or sunset (time is represented by the solar zenith angle written at the top of the Figure at the left, from Götz et al., 1934). Zenith solar sky radiation at surface is determined by ozone absorption and scattering. Zenith sky radiation at surface is progressively diminished by atmospheric ozone absorption near sunset; when the sun reaches the lowest elevation angles the scattering at higher altitudes becomes predominant which causes the reversal (Umkehr). Umkehr curves contain information on ozone profile which can be determined by a retrieval algorithm.

**Wavelengths used in total (column) ozone measurements** (see text):
*Féry spectrograph (photographic detection)*: wavelengths pairs: 306.2/326.4; 305.2/323.2; 302.2/326.4
*Dobson instrument with photoelectric detection*: 311.0/330.0
*Dobson instrument with photomultipliers*:
wavelengths pairs: A: 305.5/325.4; B: 308.8/329.1; C: 311.45/332.4; D: 317.6/339.8.
Since International Geophysical Year (IGY,1958): AD wavelengths pairs used to minimize aerosol interference.

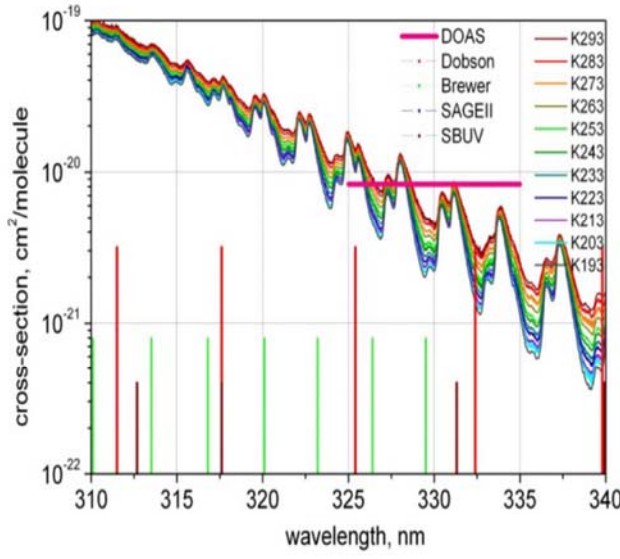

(World primary) Dobson instruments are calibrated by the Langley plot method.

Ozone absorption cross sections in the Huggins band at different temperatures and wavelengths used in different instruments (ACSO, 2015, Figure 3).

**Figure 5.** Ozone observations by instruments designed by Dobson.



During World War II, the KVV Arosa's financial support for the LKO was substantially decreased and Götz
considered leaving Switzerland. Karl Wilhelm Franz Linke, professor and director of the Institute for Meteorology
and Geophysics of the Goethe University of Frankfurt am Main (Germany) made him two offers to move to

Frankfurt. At the same time Heinrich von Ficker, professor at the University of Vienna and director of the Central Institute for Meteorology and Geodynamics, asked Götz to become professor in Vienna (Austria). However, Götz decided to stay in Arosa (in the Swiss Alps). If he had moved to Frankfurt or Vienna during World War II, the column ozone measurements made at LKO would likely have come to an end after just about one decade of measurements.

Already during the 1930s economic depression, rich clients, who had been important to some of the sanatoria, no longer could afford to travel to Switzerland. Moreover, a few years after World War II, when modern antibiotics become available, the reasons for atmospheric studies related to tuberculosis therapy at LKO gradually became obsolete (Schürer, 2017). However, starting in the 1930s, Arosa was progressively promoted as a winter sport resort area. In November 1943, Götz provided a new justification for the measurements at LKO, proposing that the excellent air quality in Arosa was a "natural resource" and that such resort areas should quantify their air quality to obtain an objective grading (Götz, 1954). This proposal was part of a project for the "medical enhancement" of Switzerland's resort areas ("Medizinischer Ausbau der Kurorte"), which was termed "climate action" ("Klimaaktion") and funded by the Swiss Federal Office for Transport. Through this project, Götz obtained support to study air pollution by making surface ozone measurements. He was convinced that high ozone concentrations were one characteristic  of healthy alpine air, since at that time the (heavily) polluted urban air had low ozone concentrations (caused by the high city-center NOx emissions titrating ozone). After World War II, Götz significantly increased efforts to obtain additional support for research at LKO by applying for a wide range of grants, which allowed him to hire collaborators who assisted him with measurements and scientific work.

In the last years of his life Götz suffered from health problems (including arteriosclerosis) (Trenkel, 1954) and he died at the age of 63 in 1954. Dr. Gertrud Perl was his main assistant from 1948 onwards and she continued making measurements even after Götz's death, but because of difficulties with Götz's wife, who owned "Villa Firnelicht" the LKO had to move to the Florentinum Sanatorium (see Fig. 2) at the end of 1953. Unfortunately, the Dobson instrument was damaged during transport to the Florentinum, so that there are a few months of data missing from the Arosa total ozone time series during this period.

## 3.    Period 1954-1962:  First intermediate period

After Götz's death, it was uncertain for several years whether the measurements at LKO would continue. Jean Lugeon, the director of MeteoSwiss (Meteorologische Zentralanstalt at the time), supported the ozone measurements at Arosa during this critical period. He knew Götz personally, since they had taught together at the University of Zürich, and was aware of the scientific value of the measurements. He was also the coordinator of the Swiss contribution to the International Geophysical Year (IGY) in 1958, in which the total ozone measurements at Arosa were recognized as a geophysically significant data set. For a few years, the Swiss National Science Foundation (SNSF) contributed to Perl's salary in addition to the support received from the KVV Arosa, the Arosa municipality and the canton Grisons. From 1957 onwards, the Arosa total ozone measurements were additionally supported by MeteoSwiss. Hans-Ulrich Dütsch, a former graduate student of Götz (see Sect. 4.1), also played an important role for the continuation of ozone measurements at Arosa. He wrote a letter to the head (minister) councilor of  the Swiss Federal Department of Home Affairs in Bern. In his response we read that MeteoSwiss

could be mandated to assume the responsibility for the Arosa ozone measurements based on several resolutions of the World Meteorological Organization (WMO), which advised that national meteorological services undertake ozone measurements. It was suggested that the Federal Meteorological Commission ("Eidgenössische Meteorologische Kommission"), the committee responsible for overseeing MeteoSwiss, should consider this in a comprehensive way, also looking at additional options, such as moving the LKO measurements to nearby Davos. Dütsch disagreed with the move to Davos, since he feared that this might lead to a serious discontinuity in the ongoing Umkehr measurements that were started in 1956 by Dütsch (see Section 4.2), because of larger aerosol contamination in Davos. In the end, the LKO stayed independent and was not integrated into MeteoSwiss, but MeteoSwiss and KVV Arosa provided financial support and measurements were continued at Arosa.

## 4. Period 1962-1985: Hans-Ulrich Dütsch

### 4.1. Dütsch and international ozone science

After Dütsch completed his PhD thesis in 1946 (title: "Photochemische Theorie des atmosphärischen Ozons unter Berücksichtigung von Nichtgleichgewichtszuständen und Luftbewegungen", Photochemical theory of atmospheric ozone under consideration of non-equilibrium states and airflow), he first worked as a physics teacher (mainly) at a high school (Gymnasium) in Zürich. However, he remained interested in ozone research and eventually decided to pursue a career in science (see Fig. 6). From 1962-1965 he lived with his family in Boulder (Colorado, USA) working as a researcher at the newly founded National Center for Atmospheric Research (NCAR). Together with Carl Mateer, Dütsch was the first to use modern computers to retrieve vertical ozone profiles with the Umkehr method.

In 1965 Dütsch was appointed as full professor at the ETH Zürich (ETHZ), where he served as director of the Laboratory of Atmospheric Physics (LAP, merged in 2001 with the Institute of Climate Sciences to become today's Institute for Atmospheric and Climate Science (IAC)). Dütsch's research continued to focus on ozone, and he continued, pursued and extended the Swiss ozone measurements (see Section 4.2).

During Dütsch's first years at ETHZ the main motivation for atmospheric ozone measurements at Arosa and Payerne was improving understanding of the "high atmosphere" circulation patterns with the aim of providing improved weather forecasts.. Publications using measurements from the nearby Hohenpeissenberg Observatory (located in Bavaria, Southern Germany) revealed links between ozone levels and synoptic weather types (Hartmannsgruber, 1973; Attmannspacher and Hartmannsgruber, 1973, 1975) and the relationship between the vertical distribution of ozone and synoptic meteorological conditions become an important research topic in the 1960s and the early 1970s (see Breiland, 1964).

Stratospheric ozone depletion resulting from anthropogenic emissions was first publicized in the 1970s. Molina and Rowland (1974) as well as Stolarski and Cicerone (1974) independently discovered that chlorine radicals destroy stratospheric ozone in a chain reaction. Furthermore, Molina and Rowland postulated that chlorofluorocarbons were a possible source gas for stratospheric chlorine. The chemical industry, particularly market leader DuPont, strongly objected to the view of Molina and Rowland. DuPont went so far as to launch an advertisement in the New York Times in 1975 stating that "Should reputable evidence show that some

fluorocarbons cause a health hazard through depletion of the ozone layer, we are prepared to stop production of the offending compounds". This provided a new justification for making high quality total ozone measurements, namely as a basis for reliable long-term trend analysis. This was a new challenge for ground-based total ozone measurements since stratospheric ozone in the extra tropics can vary by as much as ± 20 % from day to day, whereas anthropogenic stratospheric ozone changes were (and still are) on the order of only a few percent per decade.

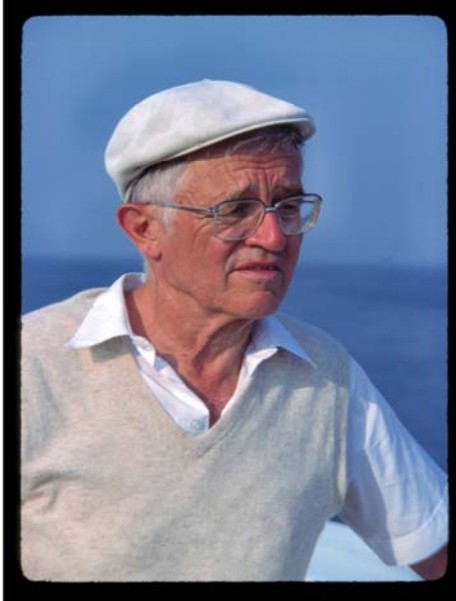

**Hans-Ulrich Dütsch**

| | |
|---|---|
| 1917 | Born on 26 Oct. in Winterthur (Switzerland) |
| | Childhood in Winterthur |
| 1940 | Diploma in theoretical physics with a minor in meteorology, University of Zürich |
| 1943-1946 | Graduate student of Götz |
| 1947-1962 | High school (Gymnasium) teacher in physics in Zurich, continuing ozone research |
| 1950 | Visiting scientist at the Massachusetts Institute of Technology, MIT, USA |
| 1962-1964 | Researcher at the High Altitude Observatory in Boulder (CO, USA) |
| | Head of the Ozone Research Program at the newly founded NCAR (CO, USA) |
| 1965-1985 | Prof. at the Swiss Federal Institute of Technology Zürich (ETH Zürich) |
| 2004 | Died on 27 Dec. in Zürich (Switzerland) |

**Figure 6.** Biography of Hans-Ulrich Dütsch.

Dütsch was one of the few scientists making important contributions to ozone research both before and after the debate on anthropogenic ozone depletion had started. Prior to this, Dütsch was largely curiosity-driven and had been interested in better understanding stratospheric ozone climatologies. For example, Dütsch (1974) provided basic science later served to validate numerical simulations of anthropogenic ozone depletion. He also contributed to the IO3C, serving first as member from 1957-1961, and then as secretary for 15 years (1961-1975), before being elected as president (1975-80), and being named an honorary member in 1984. He was also the main organizer of two important ozone symposia (the Quadrennial Ozone Symposia, organized by the IO3C) that took place in Arosa in 1961 and 1972. For more information on Dütsch's research, see also Staehelin et al. (2016.)

## 4.2    Ozone measurements at LKO under Dütsch

In 1956, Dütsch was able to find resources to ensure the Umkehr ozone measurements in Arosa continued on a regular, operational basis. When Gertrud Perl had to leave Arosa in 1962 because of health problems, Dütsch took

the responsibility and scientific leadership of the LKO, although he was still living in Boulder (CO, USA) at the time. A large majority of the observations, particularly the Umkehr measurements, were performed by students, under the tutelage of Perl and others, until Kurt Aeschbacher became responsible for the LKO measurements in 1964, remaining so until November 2001. When Dütsch became professor at ETHZ in 1965, financial support for the measurements at LKO (total ozone and Umkehr) continued as before (i.e., via KVV Arosa, Arosa municipality and the Canton Grisons). In addition to the spectrophotometric measurements, Dütsch also initiated ozone sonde measurements, which made it possible to observe ozone vertical profile in more detail. In 1966/67, these balloon measurements were operated by Dütsch from Kilchberg (close to Zürich), but in August 1968 MeteoSwiss took took over these observations and made them from Payerne, 140 km Southwest of Zürich on the Swiss plateau (Jeannet et al., 2007). In 2008 Payerne became a member of "The Global Climate Observing System (GCOS) Reference Upper-Air Network" (GRUAN) (fully certified in 2015), an international observing network under the auspices of WMO. GRUAN aims at measuring essential climate variables providing long-term, high-quality climate data records from the surface, through the troposphere, and into the stratosphere.

When Dütsch was responsible for the LKO, total ozone and Umkehr measurements were routinely performed using two Dobson spectrophotometers (see Fig. 4). To obtain the total ozone, only direct sun observations were performed. Dütsch applied the statistical Langley plot method to update the instrumental constants of the Dobson instruments every year (Dütsch, 1984). To apply the statistical Langley plot method (which was also used by Farman et al., 1985) a large number of ozone observations with different solar angles is required and therefore the observers need to choose suitable meteorological conditions, e.g. cloud free conditions lasting for at least several minutes. Each year Dütsch went to Arosa for several days to check all the total ozone measurements for reliability and to apply the statistical Langley plot method. This led to small corrections being made to the total ozone measurements for the previous year and some small changes to the instrumental constants for the following year. Students, who usually stayed in Arosa for several months at a time, made the Umkehr measurements, which need to be started prior to sunrise every morning (see Fig. 5).

In 1973, the LKO measurements were moved from the "Florentinum" to "Haus Steinbruch" (see Fig. 2), just a few hundred meters away. The working conditions at the LKO were much better at "Haus Steinbruch" than at the "Florentinum", however the running costs were higher (for more detail see Staehelin and Viatte, in prep.). In 1978, the first international intercomparison campaign of Dobson spectrophotometers took place in Arosa. This was organized by Dütsch under the auspices of the WMO. The results of this first intercomparison exercise at Arosa were not satisfying since "differences between (standard) instruments led to a debate as to which should be used as the standard for the intercomparison" (see Staehelin et al., 1998a). However, this debate deepened the insight into how necessary such comparisons were (and still are), fostering the excellent reputation of Swiss ozone research. As a result of these discrepancies Dütsch continued to apply the statistical Langley plot method to update the instrumental constants up to the begin of the 1990s.

## 5.    Period 1985-1988: Second intermediate period

### 5.1.        International development and the importance of the Arosa total ozone time series

In the early 1980s, as new information about ozone chemistry reaction rate constants became available, it seemed
that chemical ozone depletion by ODSs was considerably less than had been predicted in the late 1970s (Benedick,
1991). However, in 1985 the Antarctic ozone hole was discovered (Farman et al., 1985), and the international
ozone research community was able to demonstrate that the ozone hole was caused by the chlorine and bromine
in halocarbons, which were largely of anthropogenic origin. New insight came through the discovery that the
chlorine and bromine species are very efficiently converted into ozone destroying forms on the surface of polar
stratospheric cloud particles (Solomon et al., 1986), acting as efficient catalysts in the cold polar stratospheric
vortex (for reviews see Rowland, 1991; Peter, 1997; Solomon, 1999).
In the mid-latitudes, the first analysis based on the relatively short record of measurements from the Total Ozone
Mapping Spectrometer (TOMS) instrument onboard the Nimbus 7 satellite available at the time also showed rapid
ozone decline (Heath, 1988). However, ground-based total ozone measurements such as those made using Dobson
instruments did not confirm the large downward trends suggested by the satellite data. This discrepancy led to the
1988 publication of the International Ozone Trend Panel report (IOTP, 1988). The report demonstrated that TOMS
data available at the time were not reliable enough for trend analysis because of inappropriate treatment of the
degradation of the diffuser plate. Later these data were reanalyzed more extensively using additional wavelengths
in the retrieval algorithms and results were significantly improved (Stolarski et al., 1991). It turned out that also
some of the data from the ground-based instruments were not of high enough quality to carry out reliable long-
term trend analyses. This was attributed to calibration issues with the Dobson instruments, which showed frequent
sudden changes when compared to TOMS overpass data (IOTP, 1988). Rumen Bojkov, Secretary of the IO3C
(1984-2000), used TOMS data to provide "provisionally revised" ground based measurements, which, however,
had weaknesses such as not correcting for sulfur dioxide ($SO_2$) interferences leading to potential errors in ozone
trends based on Dobson series (e.g., De Muer and De Backer, 1992).
The most important application of the long-term measurements from Arosa (see Fig. 7) was probably their use in
the 1988 IOTP report. The Arosa time series was the only Dobson dataset that required no correction and was
much longer than any of the other ground-based measurement records. Results from Neil Harris's PhD thesis were
published in the IOTP and showed, for the first time, significant decreases in stratospheric ozone in the northern
mid-latitude winter season (Harris, 1989). He used two different approaches, namely (1) dividing the individual
records into two periods of similar length using measurements going back to 1957 and (2) developing a novel
multiple linear regression model taking into account trends for different months. In this model the downward trend
started in 1970, and the analyses also showed that the negative trend was not sensitive to the start year. At present,
standard Dobson measurements are based on observations of two (AD) wavelength pairs, which allow to minimize
the interference by aerosols, a technique introduced during the International Geophysical Year (IGY) in 1957-58
(cf. Fig. 5). To further support his main conclusion, Harris (1989) also used single other wavelengths pair (C) data
from Arosa, which are available as representative (homogenized) measurements since 1931. Again, he found
similar negative total ozone trends as at most other sites in the northern mid-latitudes (IOTP, 1988).

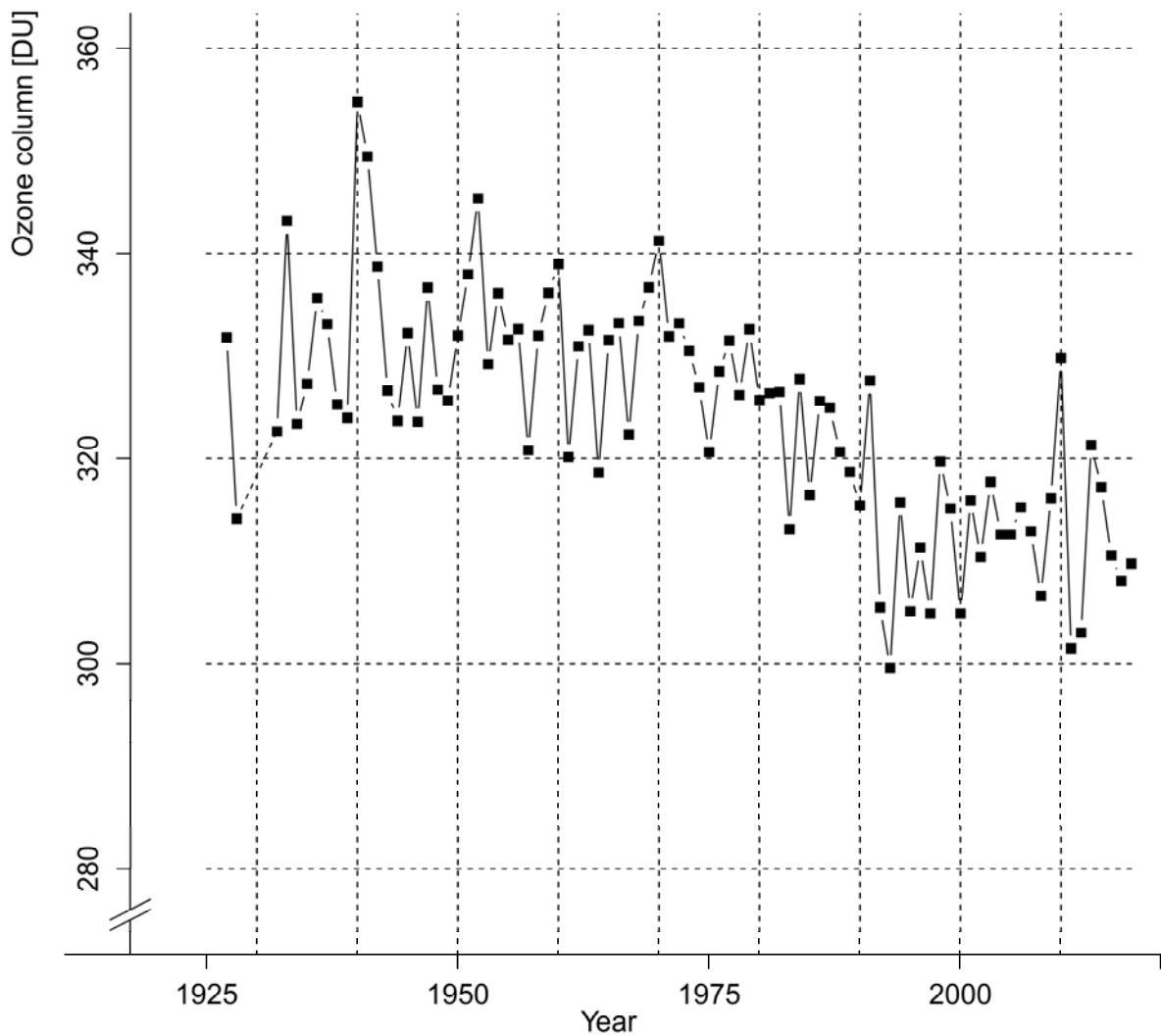

**Figure 7.** Annual mean total (column) ozone values measured at the world's longest continuous spectrophotometer site in Arosa, Switzerland, from 1926-present. The ozone column in Dobson units, where 100 DU correspond to a 1-mm thick slab of pure ozone gas at standard conditions (273.15 K, 1000 hPa).

### 5.2. Continuation of measurements at the LKO

After Dütsch's retirement in 1985, the continuation of Swiss long-term ozone measurements again became uncertain. The professor succeeding Dütsch focused on another research topic, and consequently the ETH Zürich argued that the continuation of operational ozone measurements did not fall under the responsibility of a university. Conversely, MeteoSwiss, which already was responsible for the ozonesonde measurements since 1968, argued that such long-term measurements needed scientific analysis by a well-qualified scientist, which MeteoSwiss was not able to support (a hiring freeze for permanent positions existed at the federal level at the time). Dütsch again wrote a letter to the responsible minister of the Federal government to point out the importance of the Arosa ozone measurements. Representatives from the Swiss Federal Office for the Environment (the "Swiss EPA") argued that ozone research in Switzerland needed to be continued since expert ozone researchers served a vital role to provide advice to policy makers regarding both stratospheric (in terms of the Vienna Convention and Montreal Protocol)

and tropospheric ozone. Subsequently, a commission of the Swiss academy of Natural Sciences was tasked to
analyze the situation. Government representatives as well as Swiss ozone researchers were invited to their meeting.
Again, it was considered whether it made sense to move the LKO measurements to Davos (PMOD), but no
decision was made in this regard. Nevertheless, MeteoSwiss and ETH Zürich (i.e. IAC, Institute for Atmospheric
and Climate Science since 2001, at the time Laboratory of Atmospheric Physics (LAPETH) agreed to continue the
measurements, with the former officially accepting to take responsibility for the continuation of the ozone
measurements at Arosa (total ozone and Umkehr) as well as the ozonesondes launched from Payerne, and the IAC
at ETH Zürich consenting to continue ozone research. The agreement - implying that the person responsible for
the LKO operations was moved to a MeteoSwiss position, whereas the IAC filled a scientific position with a major
focus on ozone research became effective at the beginning of 1988.
**6.    Period 1988-2014: Ozone measurements and research at MeteoSwiss and IAC (ETHZ)**

**6.1.    International Development: The Montreal Protocol**

Since 1988, the most important justification for ozone measurements at LKO Arosa (total ozone und Umkehr) and
ozone sonde launches in Payerne has been the documentation of the effect of ODSs on the stratospheric ozone
layer and the effectiveness of the Montreal Protocol. Chemical ozone depletion by ODSs is expected to evolve
very similar to the evolution of Equivalent Effective Stratospheric Chlorine (EESC). EESC provides an estimate
of the total amount of halogens in the stratosphere, calculated from emission of chlorofluorocarbon and related
halogenated compounds into the troposphere (lower atmosphere) and their efficiency in contributing to
stratospheric ozone depletion (hence "effective"), and by taking the higher ozone destructiveness of bromine
appropriately into account (hence "equivalent"). EESC peaked in the second half of the 1990s and subsequently
showed a slow decrease, which is attributable to the Montreal Protocol, but in its slowness dictated by the long
lifetimes of the emitted substances (see Fig. 8a). Total ozone measurements at Arosa are broadly consistent with
long-term evolution of EESC (Staehelin et al., 2016) showing record low values in the early 1990s (Fig. 8b, cf.
Fig 7). The recovery of the ozone layer is a slow process and the signal of any sort of turnaround in the Arosa total
ozone time series is still indistinct. Figure 8b shows the large interannual variability of the annual means, which is
normal for a single measurement station and renders an attribution of the change in the downward trend difficult.
While model results suggest that the Montreal Protocol and its amendments and adjustments have helped to avoid
millions of additional skin cancer cases, Fig. 8b indicates that the global network of ozone station measurements
needs to remain strong in order to achieve a clear detection of the trend reversal and a proper attribution of the
reasons.



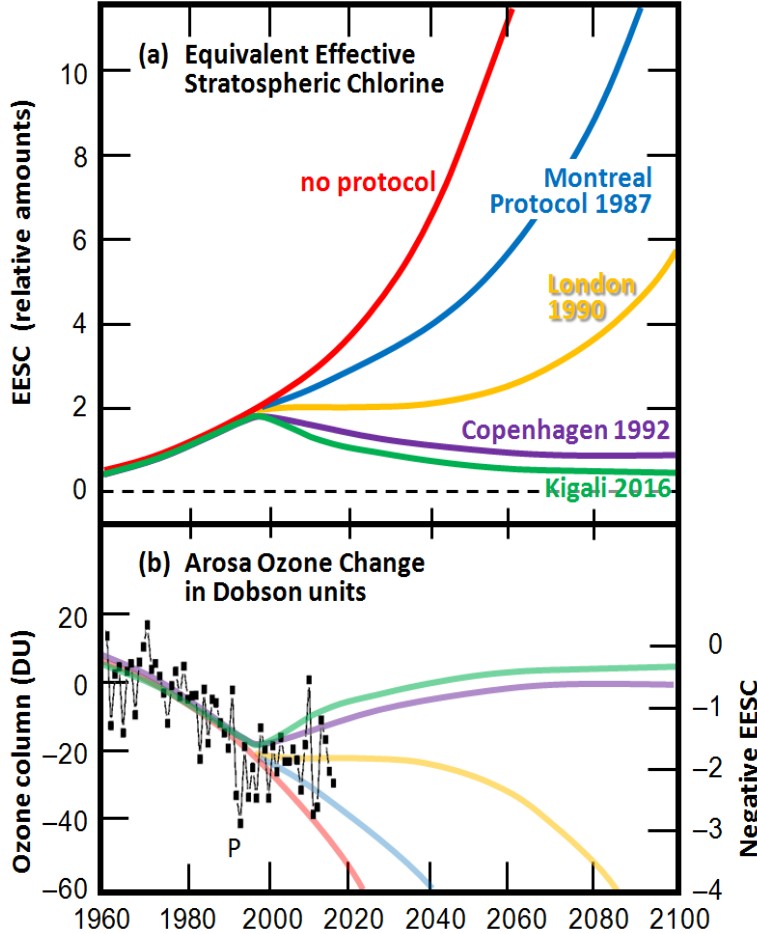

**Figure 8.** (a) Relative abundance of Ozone Depleting Substances (ODSs, i.e. volatile halocarbons) expressed as equivalent effective stratospheric chlorine (EESC) for the mid-latitude stratosphere, shown for various scenarios (demonstrating the impact of the Montreal Protocol and its subsequent Adjustments and Amendments). EESC can be viewed as a measure of chemical ozone depletion by ODSs and takes into account the temporal emission of the individual ODS species as well as their ozone depleting potential. (b) Arosa annual mean ozone columns (black symbols, as in Fig. 7) in comparison with the scenarios in (a). "P" marks the eruption of Mt. Pinatubo in 1991, which has aggravated the ozone loss.

## 6.2  LKO and related activities

### 6.2.1 Cooperation between MeteoSwiss and IAC (ETHZ)

The cooperation between MeteoSwiss and the IAC of ETH Zürich ensured that the different strengths of the two institutions were fully utilized. MeteoSwiss had the expertise and resources to renew the infrastructure at the Arosa station and was also able to guarantee reliable long-term operation through permanent contracts for technicians and scientists. On the other hand, IAC (ETH Zürich) had the possibility to lead scientific research, for example, with PhD theses that produced results published in the scientific literature. The use of ozone measurements as basis for scientific research requires high quality data and the results from the ETH studies thus provided both, a feedback mechanism in terms of data quality and enhanced visibility of the ozone measurements.

### 6.2.2 Renewal of the LKO infrastructure

When Meteoswiss become responsible for the LKO ozone measurements in 1988, the instrument infrastructure required renewal and extension. This was completed under the leadership of Bruno Hoegger and included constructing a spectrodome to house the two Dobson spectrophotometers as well as semi-automation of the Dobson total ozone measurements and full automation of the Dobson Umkehr measurements (Hoegger et al., 1992). Three Brewer instruments were also purchased between 1988 and 1998, thus allowing increased reliability of the Arosa total ozone series by complementing the Dobson Umkehr measurements and by providing instrumental redundancy (see Fig. 4). Furthermore, UVB measurements were added. For more technical information including new electronics see Staehelin and Viatte, in prep. Stübi et al. (2017a) demonstrated the excellent stability of the Arosa Brewer triad over the past 15 years.

### 6.2.3 Homogenization of the Arosa total ozone and Umkehr timeseries

The Dobson instrument D15 was the main instrument used to measure total ozone in Arosa from 1949 to 1992 (see Fig. 4). Archie Asbridge (formerly of Atmospheric Environment Canada) inspected this instrument after it was taken out of service in 1992, and it turned out that it had been operated in optical misalignment. Using the overlap between total ozone measurements of the D15 and D101 instruments, the latter of which was calibrated against the world standard instrument in 1986 and again in 1990, the Arosa column ozone time series was adjusted to the scale of the world primary Dobson instrument (for more detail see Staehelin et al., 1998a and Scarnato et al., 2010). The Arosa Umkehr time series also required homogenization (Zanis et al., 2006).

### 6.2.4 Foci of scientific studies since the 1990s

The comparison of the unique Arosa total ozone time series from Dobson and Brewer instruments has allowed studies of the differences between the two instrument types (Staehelin et al., 1998a; Scarnato et al., 2009, 2010) as well as their long-term behavior since they are calibrated in different networks. The large data set of quasi-simultaneous measurements was particularly valuable for studying the effect of temperature dependence of ozone absorption cross-sections on total ozone measurements attributable to the different wavelengths used in Dobson and Brewer instruments (Scarnato et al., 2009, Redondas et al., 2014). These results were an important contribution to the GAW ACSO (Absorption Cross-Sections of Ozone) project in which available laboratory cross-sections of atmospheric ozone measurements were studied (ACSO, 2015; Orphal et al., 2016).

In the 1990s, quantification of the downward ozone trends was the main reason for making long-term stratospheric
measurements (comp. Section 5.1, and Staehelin et al., 1998b, 2001). These trends were seen as a consequence of
increasing ODS concentrations. Subsequent studies were also devoted to understanding the potential contribution
of other processes enhancing the observed downward trends, including long-term climate variability, e.g. related
to tropopause altitude (Steinbrecht et al., 1998) and climate patterns (Steinbrecht et al., 2001). The unique length
of the Arosa total ozone series was very valuable in demonstrating that the North Atlantic Oscillation (NAO) or
Arctic Oscillation (AO) enhanced downward winter ozone trends in central Europe for the period up to the
mid1990s (Appenzeller et al., 2000; Weiss et al., 2001). Brönnimann et al. (2004a, 2004b) also showed that the
record high values of total ozone at Arosa that occurred in the early 1940s were due to an increase in strength of
Brewer Dobson circulation caused by a very large ElNino/Southern Oscillation anomaly during that period.
The unique length and high quality of the Arosa total ozone and Umkehr measurements also meant they were
important for the EU project CANDIDOZ (Chemical and Dynamical Influences on Decadal Ozone Change; Zanis
et al., 2006; Brunner et al., 2006; Harris et al., 2008). Later, as the ODS concentrations have decreased,
documentation of the "turn around" in stratospheric ozone trends became more and more important (e.g. Mäder et
al., 2010). The Arosa time series was also used to introduce the concept of extreme value theory in ozone science
(Rieder et al., 2010a, b). This allowed attribution of extreme ozone values to events of various origins, dynamical
features such as ENSO or NAO, or chemical factors, such as cold Arctic vortex ozone losses, or major volcanic
eruptions of the 20[th] century, e.g. Mt. Pinatubo.

**6.2.5 Tropospheric ozone**
The surface ozone measurements from Arosa are unique and very valuable for tropospheric chemistry studies.
Surface ozone measurements were begun already in the 1930s by Götz to quantify the contribution of tropospheric
ozone to the total column, and were later continued by the careful and representative surface ozone measurements
made in the 1950s (Götz and Volz, 1951; Perl, 1961). Thanks to these measurements it was possible to show that
surface ozone concentrations increased by more than a factor of two from the 1950s to 1990 (Staehelin et al.,
1994). This has commonly been attributed to the large increase in ozone precursor emissions (nitrogen oxides,
volatile hydrocarbons, and carbon monoxide) resulting from the strong economic growth in industrialized
countries following World War II. The surface ozone measurements made at Arosa and Jungfraujoch were pillars
in the studies of Parrish et al., (2012, 2013), which contributed to an important report by the Task Force of the
Hemispheric Transport of Air Pollution (HTAP). HTAP was organized in 2005 under the auspices of the United
Nations Economic Commission for Europe (UNECE) Convention on Long-range Transboundary Air Pollution
(LRTAP Convention) to study intercontinental transport of ozone in northern mid-latitudes. Based on these data,
Parrish et al. (2014) compared three state-of-the-art chemistry climate models (CCMs) to show that simulated
surface (baseline) ozone trends over Europe were about a factor two smaller than those seen in the available
observations. This result was recently confirmed by Staehelin et al., 2017.
**7.       Future of ozone measurements at the LKO**
7.1       **International Demands**

Policy makers and the general public would like to see proofs of the effectiveness of the Montreal Protocol and to
better understand how climate change will affect the ozone layer, i.e. what are the impacts of the stratospheric
cooling and the anticipated enhanced Brewer-Dobson circulation on ozone, and what this means for polar, mid-
latitude and tropical ozone.
Recovery of the stratospheric ozone layer in response to the reduction of ODS concentrations controlled by the
Montreal Protocol is slow (see Sect. 6.1) and requires continued long-term stratospheric ozone observations. ODSs
most directly impact ozone in the upper stratosphere, where photolysis leads to the release of halogen radicals
from these species. Extensive data analyses carried out under the auspices of the SI2N activity commonly
sponsored by SPARC (Stratosphere-troposphere Processes and their Role on Climate), IO3C, IGACO-O3/UV
(Integrated Global Atmospheric Composition Changes), and NDACC (Network for Detection of Atmospheric
Composition Changes) highlighted issues related to the availability and uncertainty of measurements. Recent
examples are merged satellite datasets, and trend analysis techniques (see the special journal issue jointly organized
between Atmospheric Chemistry and Physics, Atmospheric Measurement Techniques, and Earth System Science
Data: Changes in the vertical distribution of ozone – the SI2N report). Steinbrecht et al. (2017) presented a recent
analysis of upper stratospheric ozone trends confirming the expected increase in upper stratospheric ozone in
extratropics. Finally, Ball et al (2018) showed that total ozone in the mid-latitudes has not increased as expected
and their careful analysis of mostly satellite measurements indicated a downward trend in the lower stratosphere
(15-22 km) which continued since 1987. The physical cause of this surprising trend is presently unknown and
requires further study.
It is vital to continue high quality stratospheric ozone measurements to be able to follow the slow recovery of the
ozone layer in response to the changing burden of stratospheric ODSs, including nitrous oxide ($N_2O$), which is
likely to become the dominant species for stratospheric ozone depletion in future (Ravishankara et al., 2009;
Portmann et al., 2012).
Climate change will modify the distribution of stratospheric ozone in different ways (see e.g. Arblaster et al.,
2014). Increasing greenhouse gases cause decreasing stratospheric temperatures, which in turn modify reaction
rates and lead to increasing extra-tropical stratospheric ozone concentrations. This is not the case over the poles,
where the stratosphere is not expected to cool on average. Furthermore climate change is expected to enhance the
Brewer Dobson Circulation which transports ozone from the main tropical production region to the extra-tropics
(Butchart, 2014). Modification of the Brewer Dobson Circulation is expected to increase stratospheric ozone in
the mid-latitudes to levels above those seen in the past; tthis has been termed "super recovery". In contrast, the
enhanced transport out of the tropics is expected to result in a decrease in stratospheric ozone in these regions. The
enhancement of the Brewer Dobson Circulation is, however, still under debate, with state-of-the-art CCMs
projecting an increase but only controversial observational evidence being available. Importantly, the expected
enhancement depends strongly on the climate change scenario investigated, thus it is essential that high quality
measurements are continued.
The unique length of the Arosa timeseries is particularly useful for documenting the effects of climate change on
ozone since the dataset covers a period of almost 40 years when the stratosphere was relatively undisturbed by
anthropogenic influence, about 25 years in which anthropogenic ODSs increased in concentration in the
stratosphere, and the latest period with the slow decrease in stratospheric ODS concentrations. The Arosa
timeseries will therefore play a crucial role in the coming decades to further document ozone changes in the
Northern mid-latitudes, including the predicted "super recovery" expected to become important around 2030 (e.g.
Hegglin et al., 2015).

**7.2 Continuation of measurements at the LKO**

The MeteoSwiss board of directors decided in 2015 to explore the possibility of moving the Arosa measurements
to the PMOD in Davos. Such a move would result in reduced measurement costs in combination with the advantage
of the excellent technical infrastructure and expertise that is available at the PMOD in Davos. Within this activity
the Dobson instruments are currently completely automated (comp. Fig. 4). However, before such a move is to
take place, a multiannual period of overlapping measurements at both sites (Arosa and Davos) is essential. A break
in the world's longest total ozone time series would be very unfortunate. A relocation is particularly challenging
as stratospheric recovery from ODS is expected to be slow (see Sec. 6.1) meaning ozone changes will be small
and thus very high quality (i.e. very high stability) measurements are required. At present simultaneous total ozone
measurements of Brewer instruments of Davos and Arosa have been analyzed and presented (Stübi et al., 2017b).
**8. Summary and Conclusions**

Homogenous long-term records such as the total ozone record from Arosa are very valuable for trend analyses in
climate science. Reliable long-term, ground-based total ozone measurements are also crucial for validation of
ozone observations from space, particularly in terms of validating the long-term stability of merged satellite
datasets (e.g. Labow et al., 2013). Furthermore, they serve as a baseline for evaluating numerical simulations such
as Chemistry Climate models (CCMs), which are used to make projections of future ozone evolution (see e.g.
Eyring et al., 2013, Arblaster et al., 2014). The extraordinary length of the Arosa record was important for a wide
range of studies, including the analysis of stratospheric ozone related to long-term climate variability such as the
NAO/AO (Appenzeller et al., 2000) and El Nino Southern Oscillation (Brönnimann et al., 2004a and 2004b).
Furthermore, the measurements have been very valuable for the evaluation of the (early part of the) Twentieth
Century Reanalysis Project (Compo et al., 2011; Brönnimann and Compo, 2012).
The reasons for continuing the Arosa measurements have changed many times over past decades, and it was  never
imagined that such a long record could be established. Fig. 9 provides a historical overview of international ozone
research in connection with the different phases of the LKO, which also indicates various funding periods. The
justification for the LKO measurements for society can be summarized as

(1) to study environmental factors potentially important for the medical recovery from pulmonary TB
(relevant from the beginning until around World War II),
(2) to investigate air quality as an important natural resource in resort areas (as discussed in the second half
of World War II)
(3) to improve our understanding of atmospheric physics for improved weather forecasts (important in the
1960s and early 1970s)
(4)  to quantify anthropogenic ozone destruction by ODSs (mid-1970s to mid-1990s)
(5)  to document the effectiveness of the Montreal Protocol in saving ozone (since around the middle of the

1990s)

(6)  to understande the mutual relationship between climate change and global ozone depletion, and the

effectiveness of the Montreal protocol (this century)


**Figure 9.** Historical overview of the successive periods of Light Climatic Observatory of Arosa (LKO). Total
ozone measurements (top, annual means); different phases during the history of LKO including main sponsors (in
orange), justification of measurements for society (in yellow); milestones in international ozone research, and
international legislation (blue).

A key element for the success of LKO measurements and its continuation was the motivation of the scientists
involved, i.e. Götz's early initiative and Dütsch's persistence.
From our experience the following issues were most relevant for the successful operation of LKO over the last
decades:
-   Redundancy allows for increased credibility of measurements, which is particularly important for reliable

long-term trend analysis. At Arosa, 3 Dobson and 3 Brewer spectrophotometers were simultaneously

operated since 1998, which helps to obtain important scientific results regarding Dobson and Brewer

spectrophotometers relevant within the broader context of atmospheric ozone measurements.

-     Regular comparison of station instruments with standard spectrophotometers operated under the WMO
umbrella are important for high-quality measurements and consistency of ozone measurements within a
particular network.

-     Scientific analysis and use of stratospheric ozone measurements in scientific publications and model
intercomparisons not only enhances visibility of the measurements within the community, but also is a
quality assessment, which might motivate scientists and technicians operating the measurements.

-     Reliable techniques are important for high quality stratospheric ozone measurements including
automation to reduce manpower costs and to make measurements less dependent on the skills of an
individual operator.

It is difficult to obtain funding for continuous observations through normal science funding agencies such as the
Swiss National Science Foundation (SNSF), since an additional few years of measurements usually do not result
in novel scientific conclusions. This is the experience within other networks as well, for example NDACC. The
success of the Montreal Protocol measures probably contributed to the decrease in the number of ozone
measurements submitted to the World Ozone and Ultraviolet Data Center (WOUDC, presently operated by
Environment and Climate Change Canada) over the past few years (Geir Braathen, personal communication). This
might be exacerbated in the future as monitoring costs come under further pressure in many countries. However,
we believe that such routine measurements are the responsibility of developed countries. Institutions like national
meteorological services, although they also may experience financial shortfalls, are ideally suited to carry out these
types of measurements since they are (in contrast to universities) capable of making long-term commitments and
have the possibility to hire permanent staff. On the other hand, universities have the advantage of being able to
focus on particular issues (e.g. through PhD theses) for a limited time, resulting in articles in peer-reviewed
journals. It is important to stress the relevance of scientific activities using long-term observations. Excellent
collaboration has existed between MeteoSwiss and the IAC (ETHZ) for the past three decades. However, this
particular type of cooperation will be less feasible in future, as the required permanent scientific positions will
typically no longer be available at universities. In other countries the research aspects are often integrated in the
same institution (e.g. the German Weather Service (DWD) in Germany or the "Centre National de la Recherche
Scientifique (CNRS)" in France). This problem still awaits a proper solution for the Swiss long-term ozone
measurements.
From the very beginning, the ozone measurements from Arosa (initiated by the fruitful collaboration between Götz
and Dobson) have been an important contribution both to the global network of ozone measurements and to ozone
research. During the early part of the record, the International Ozone Commission (IO3C) of IAMAS coordinated
the ozone measurements. Since the 1970s WMO has taken the lead, first in the framework of the Global Ozone
Observing System (GOO3S), later the Global Atmosphere Watch (GAW) programme (SAG-ozone) became
responsible for overseeing and coordinating stratospheric ozone measurements to obtain and maintain high quality
data suitable for long-term trend analysis. GAW might continue these activities in collaboration with other
networks, such as NDACC, the present Brewer COST network, and the IO3C in order to (i) maintain and extend
high quality records of ground-based ozone stations and (ii) to continue comparisons of Dobson and Brewer
measurements with other/new instruments such as SAOZ and PANDORA. GAW might represent the ground-
based community as partners to the satellite community, for example within the Copernicus project and GAW also
can contribute to research programs and initiatives, illustrated by the long history of ozone research connected
with the LKO started by the pioneers Götz and Dütsch and continued more recently by MeteoSwiss and ETHZ
under the auspices of WMO, IGACO-O3/UV, ACSO, and SPARC.
Beyond any doubt the Montreal Protocol (including enforcements) has been very successful for the protection of
the ozone layer over densely populated areas, avoiding large damage by manmade chemicals as shown by extended
numerical simulations (Newmann et al., 2009). In the future, when the stratosphere is expected to gradually recover
from the decreasing burden of ODSs, continued observations will not only be required to document the expected
increase in stratospheric ozone, but also to document the effects of climate change on stratospheric ozone, as
predicted to happen by CCMs, i.e. through enhancement of the Brewer Dobson Circulation and possible other
effects connected with climate change (Ball et al., 2018).
*Acknowledgements.* Several present and former colleagues from MeteoSwiss contributed to the study of the history
of the LKO namely, Dr. Bruno Hoegger, Kurt Aeschbacher, and Herbert Schill. We acknowledge the help of
Renzo Semadeni (Kulturarchiv Arosa-Schanfigg), Peter Bollier (retired teacher in history in the Alpine
Mittelschule Davos and expert in the history of Fridericianum), Dr. Hans Ulrich Pfister (Staatsarchiv des Kantons
Zürich), Susanne Wernli (Gemeindeverwaltung Davos), Simon Rageth and Florian Ambauen (Rhätische Bahn
AG), Klaus Pleyer (German Sanatorium (Deutsche Heilstätte, today Hochgebirgsklinik Davos), Roesli
Aeschbacher (wife of Kurt Aeschbacher), and several colleagues from the Swiss Federal Archives. Finally, we
would like to thank Prof. Johannes Gartmann (medical director of Sanatorium Altein (1958-78) for valuable
discussion, Dr. Wolfgang Steinbrecht (Observatorium Hohenpeissenberg of the German Weather Service (DWD)
for helping to find literature related to ozone measurements and synoptic meteorology, and Bob Evans (formerly
at NOAA, Boulder, USA) who supplied us with some information about the Dobson instruments operated at Arosa.
We also want to thank to Rachel Vondach for drawing Figure 2.

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
