# Peer review of "Stratospheric ozone measurements at Arosa (Switzerland)"

_Atmospheric Chemistry and Physics, 2017_

## Referee Comment (RC1) · Anonymous Referee #1 · 27 Dec 2017

This is the first review of the paper by Staehelin et al. titled "Stratospheric ozone measurements at Arosa (Switzerland): History and scientific relevance". The manuscript depicts the history of Arosa's longest ozone record and describes different periods in the record. It is very important to understand the circumstances that brought public interest and financial support for the ozone research at Arosa. From the very beginning, the need for the continuous surface and total column ozone measurements was publicized to benefit public health (i.e. studies of the tuberculosis, UV-exposure related cancer, pollution, etc.). The importance of continuous measurements has been recognized due to short term (meteorological) and inter-annual (Dobson-Brewer circulation) variability that impacts ozone measurements. The laboratory studies of the CFCs in 1970-1980s and their link to the destruction of stratospheric ozone required verifica-

tion by observations. It was possible through analyses of very long and stable record that provided information on ozone depletion detected in the 1990s. Researchers used Arosa record for ozone trend analyses and attribution of changes to anthropogenic activities, volcanic aerosols and climate drivers. The manuscript also outlines management approach that allowed Arosa research to continue with data collection during multiple potential interruptions for funding sources. This is very important lesson to current observational programs struggling with financial support. There are lessons learned on how to create successful project through collaboration between different organizations, including perseverance of scientists reaching out to the state government programs and private sector, and using scientific publications to emphasize the importance of the continued ozone record. Acknowledgement of the importance of ozone observations by international bodies of the World Meteorological organization and the Parties of the Montreal Protocol has been indeed very important tool used by observatories to seek continuous observation funding. There is also a concern about future move of Dobson instruments to Davos. This is not the first time when the merger of the Arosa and Davos ozone observing programs was proposed. In the past, it was possible to argue about importance of the continuous record and impacts of the local meteorological and anthropogenic influences on the record that can interfere with monitoring of ozone recovery. The studies of comparability of the records between Brewer measurements at two locations has been published that argue that both locations show little difference. However, it is still important to validate the move of the historical Dobson instruments to Davos. It is very important to provide information of the move to scientific community and engage scientists in the research of the impacts of location on continuous longest ozone record in the World. Here are my suggestion for the paper. 1) It will be nice to create the graph that would show the time line that combines information about the subject of the research at Arosa with funding information, and also includes important research publications dates in the time line. It may be possible to add this information in Figure 4 (Figure needs to be enlarged for final publication – too small font and hard to read). 2) In Conclusion section, it would be nice to have "Lessons Learned"

for managing of the station operations: few bullets about actions that helped to organize data gathering and research. Detailed comments: p. 5 line 143-144 "After the first conjectures that the amount of biologically active UV-radiation was determined by stratospheric ozone levels" – is there a reference for the paper? Is it Gotz, 1926 p.10, line 297, please provide the year when Payerne joined the GRUAN program. p.10, lines 297-298, "To apply the statistical Langley plot method a large number of ozone observations with different solar angles is required and . . ." – Was Langley plot method applied to other Dobson instruments outside of Arosa? What is the limitation of the Langley method and why it was not successful at other stations? p. 15, line 438-440. Please provide information about Brewer automation, new electronics and less stratospheric temperature sensitivity in ozone retrievals. p. 16, lines 452-469. This section needs to be expanded to provide more information on how Arosa data contributed to the AO, NAO and ENSO studies. It is mentioned very briefly now, but it is very important contribution. p. 16, line 484, since this paper discusses the value of the long-term observations, it would be good to state that the observations provides reference for validation and improvement of the models. p. 18, Summary and Conclusion: It would be nice to organize this section to more clearly emphasize benefits of Arosa record to the ozone science. You already have statements in regards to validation of satellite stability, consistency in the merged satellite records. I wonder if it can be separated into individual paragraphs that can read as an expanded bullets. p. 18, lines 549-553. I suggest adding the years of Arosa record to each of the justifications. It will show reader how sometimes focus of the society comes back in later years when more theoretical information and observations become available to renew the subject of the early studies. p.18, lines 553-555 – this needs to be a separate paragraph concentrating on the need for future ozone measurements. The validation of future satellite stability, resolving differences between satellite records, and help to bridge possible breaks in the satellite records can be mentioned here. As well as continuous validation of the CCM, CTM and regional model, can be added to this paragraph. Mentioning of the Copernicus project is also important. Help with validation of the new ozone measuring

instruments is one of the Dobson network objectives. Providing training for future scientists, including training of operators from developing countries must be mentioned in this section. p. 18 lines 555-559 – this is very important section on the management of the funding for station observations, including personal communications of scientists with funding agencies. This needs to be emphasized in the "lessons learned" . p. 18, line 565 – remove second "cost" p18, lines 560-566, p. 19 lines 567-578. It is important to mention guiding role of the WMO ozone SAG committee and Ozone research managers' report that set goals for the WMO Dobson network. It seems that this venue should be used more often than not to support the funding needs for individual countries. You should also emphasize that Arosa record on its own would not be able to succeed or become widely used without collaborations with the WMO Dobson network and NDACC. The recent efforts in the Brewer COST action project are aimed at creating the organized collaborative network of ozone measurements in Europe and homogenization of the near-real time processing of ozone data from multiple Brewer instruments. The effort to retain all individual station data on the same scale was the most important step in creating the homogenized operations for the Dobson network in 1980s.

---

## Referee Comment (RC2) · Anonymous Referee #2 · 22 Jan 2018

[referee-annotated manuscript omitted]

---

## Author Comment (AC2) · 8 Mar 2018

We would like to thank Referee 2 for his/her careful review and valuable suggestions. The comments of the reviewers are followed by our replies. I found the paper very interesting, as I'm not a specialist in stratospheric ozone so historical details of how some of the researched developed was new to me. Overall the paper is well written and I only saw a couple of minor grammatical errors. It gives a detailed description of how total column ozone measurements were initiated at Arosa with the aim of studying the impact of mountain air on recovery from tuberculosis and eventually contributed to our understanding of the stratospheric ozone layer and the damage being caused to it by our emissions. The story it tells of pressures from competing scientists and institutes or the difficulties in maintaining funding, illustrate the ways in which science develops

in the face of many challenges. It highlights the importance of maintaining high-quality and long-term measurement facilities as the information they give can evolve with time as we become aware of new processes the consequences of human activity. The site is of continued importance for studying the impact of climate change on ozone as well as other factors. My only suggestion would be that the abstract could possibly put a greater emphasis on the value of the historical measurements and continuing them into the future, in the light of climate change. Our reply: We tried to adapt the Abstract accordingly Grammatical notes are in the attached pdf. Our reply: We changed the manuscript accordingly. We also improved some formulations. We thought about the remark of the referee: Could you add a couple of sentences to outline the basic operation of these types of instrument and why the Umkehr was such a significant improvement? Not all readers may be familiar with the optical methods used to measure total column ozone. Our reply: In Section 2.3, third paragraph we slightly extended the text and we added a new Figure (Figure 5 in the revised manuscript) to provide a short summary of Dobson spectrophotometry (total ozone as well as Umkehr) for those readers not familiar with these instruments. However, we found that adding more text is more problematic because the paper is structured following the historical evolution of the institution and does not contain a section describing methods.

---

## Author Response (AR1)

**Reply to Referee 1**

We would like to thank Referee 1 for his/her very competent and careful review and the valuable suggestions. We first show the comments of the reviewers, following our replies.

*This is the first review of the paper by Staehelin et al. titled "Stratospheric ozone measurements at Arosa (Switzerland): History and scientific relevance". The manuscript depicts the history of Arosa's longest ozone record and describes different periods in the record. It is very important to understand the circumstances that brought public interest and financial support for the ozone research at Arosa. From the very beginning, the need for the continuous surface and total column ozone measurements was publicized to benefit public health (i.e. studies of the tuberculosis, UV-exposure related cancer, pollution, etc.). The importance of continuous measurements has been recognized due to short term (meteorological) and inter-annual (Dobson-Brewer circulation) variability that impacts ozone measurements. The laboratory studies of the CFCs in 1970-1980s and their link to the destruction of stratospheric ozone required verification by observations. It was possible through analyses of very long and stable record that provided information on ozone depletion detected in the 1990s. Researchers used Arosa record for ozone trend analyses and attribution of changes to anthropogenic activities, volcanic aerosols and climate drivers. The manuscript also outlines management approach that allowed Arosa research to continue with data collection during multiple potential interruptions for funding sources. This is very important lesson to current observational programs struggling with financial support. There are lessons learned on how to create successful project through collaboration between different organizations, including perseverance of scientists reaching out to the state government programs and private sector, and using scientific publications to emphasize the importance of the continued ozone record. Acknowledgement of the importance of ozone observations by international bodies of the World Meteorological organization and the Parties of the Montreal Protocol has been indeed very important tool used by observatories to seek continuous observation funding. There is also a concern about future move of Dobson instruments to Davos. This is not the first time when the merger of the Arosa and Davos ozone observing programs was proposed. In the past, it was possible to argue about importance of the continuous record and impacts of the local meteorological and anthropogenic influences on the record that can interfere with monitoring of ozone recovery. The studies of comparability of the records between Brewer measurements at two locations has been published that argue that both locations show little difference. However, it is still important to validate the move of the historical Dobson instruments to Davos. It is very important to provide information of the move to scientific community and engage scientists in the research of the impacts of location on continuous longest ozone record in the World.*

*Here are my suggestion for the paper.*
1) *It will be nice to create the graph that would show the time line that combines information about the subject of the research at Arosa with funding information, and also includes important research publications dates in the time line. It may be possible to add this information in Figure 4 (Figure needs to be enlarged for final publication – too small font and hard to read).*

Our reply: We followed the idea of the referee and produced an overview Figure, Figure 9 in the revised manuscript. We took the data of Fig. 6 instead of Fig. 4 (originally submitted manuscript) as proposed by the referee.

2) *In Conclusion section, it would be nice to have "Lessons Learned" for managing of the station operations: few bullets about actions that helped to organize data gathering and research.*

Our reply: We followed the advice of the referee summarizing the most important points to run the station in bullets points (see Section 9 (Summary and Conclusions), 4. paragraph in the revised manuscript).

*Detailed comments:*

*p. 5 line 143-144 "After the first conjectures that the amount of biologically active UV-radiation was determined by stratospheric ozone levels" – is there a reference for the paper? Is it Götz, 1926.*

Our reply: The title of the paper Götz 1926 shows that Götz was aware that ozone is responsible for the transparency of the atmosphere for solar UV-B radiation. This relationship was first discussed by Hartley (1871). However, we wanted to emphasize that Götz who was interested to understand UV radiation at the Earth's surface started to study stratospheric ozone because he was not satisfied by measurement of UV-climatology. In a presentation to the Royal Institute of Public Health in May 1929 he explains his scientific approach (which was obviously different to the view of Dorno from Davos): "From the beginning the measurements of the UV light in Arosa are not only designed on a statistical basis, but also driven by the desire to understand the variability. And the observed higher UV-levels in the autumn with respect to the spring observed in the series 1921 to 1924 can only be explained by the weaker ozone layer in the autumn. The variations of the ozone layer which has first been observed quantitatively by Fabry and Buisson are decisive for the comprehension of the behavior of the UV-Radiation which is essential for the life" (for more detail see Staehelin and Viatte, in prep.). We modified the manuscript, see Section 2.3, 2. paragraph.

*p.10, line 297, please provide the year when Payerne joined the GRUAN program.*

Member since 2008, (fully) certified in 2015 (see revised manuscript, Section 4.2, 1. paragraph)

*p.10, lines 297-298, "To apply the statistical Langley plot method a large number of ozone observations with different solar angles is required and .." – Was Langley plot method applied to other Dobson instruments outside of Arosa? What is the limitation of the Langley method and why it was not successful at other stations?*

Our reply: The statistical Langely plot method was used in the Halley Bay total ozone record of the British Antarctic survey, which was used to discover the ozone hole over Antarctica (Farman et al., 1985, see Section 4.2, second paragraph in the revised manuscript). The application of the statistical Langley plot method requires a large number of individual measurements covering an extended range of my-values of a large number of days and the application of the method is time consuming. I think the requirement of a large number of measurements and the rather time consuming data analysis are the reason why the method is not commonly used as well as the regular intercomparisons organized under the auspices of WMO.

*p. 15, line 438-440. Please provide information about*

Our reply: For technical information concerning new electronics and use of Brewer spectrophotometers at Arosa we refer to the report of Staehelin and Viatte, which is in prep. whereas the use of Arosa data to study temperature sensitivity is discussed in the first paragraph in 6.2.4 in the revised manuscript.

*p. 16, lines 452-469. This section needs to be expanded to provide more information on how Arosa data contributed to the AO, NAO and ENSO studies. It is mentioned very briefly now, but it is very important contribution.*

Our reply: We extended the discussion in the revised manuscript.

*p. 16, line 484, since this paper discusses the value of the long-term observations, it would be good to state that the observations provides reference for validation and improvement of the models.*

Our reply: We added Eyring et al., 2013 as reference

*p. 18, Summary and Conclusion: It would be nice to organize this section to more clearly emphasize benefits of Arosa record to the ozone science. You already have statements in regards to validation of satellite stability, consistency in the merged satellite records. I wonder if it can be separated into individual paragraphs that can read as an expanded bullets.*

Our reply: We summarized our experience from the last decades in form of bullet points (see above)

*p. 18, lines 549-553. I suggest adding the years of Arosa record to each of the justifications. It will show reader how sometimes focus of the society comes back in later years when more theoretical information and observations become available to renew the subject of the early studies.*

Our reply: We modified the manuscript accordingly (Section 8, second paragraph).

*p.18, lines 553-555 – this needs to be a separate paragraph concentrating on the need for future ozone measurements. The validation of future satellite stability, resolving differences between satellite*

*records, and help to bridge possible breaks in the satellite records can be mentioned here. As well as continuous validation of the CCM, CTM and regional model, can be added to this paragraph.*

Our reply: The points related to the future including validation of numerical simulation are discussed in the second paragraph of Section 8 in the revised manuscript.

*Mentioning of the Copernicus project is also important. Help with validation of the new ozone measuring instruments is one of the Dobson network objectives. Providing training for future scientists, including training of operators from developing countries must be mentioned in this section.*

Our reply: These points are addressed in the last paragraph of Section 8.

*p. 18 lines 555-559 – this is very important section on the management of the funding for station observations, including personal communications of scientists with funding agencies. This needs to be emphasized in the "lessons learned" .*

Our reply: This discussion is covered by 5. paragraph of Section 8.

*p. 18, line 565 – remove second "cost".*

Our reply: Done

*p18, lines 560-566, p. 19 lines 567-578. It is important to mention guiding role of the WMO ozone SAG committee and Ozone research managers' report that set goals for the WMO Dobson network. It seems that this venue should be used more often than not to support the funding needs for individual countries. You should also emphasize that Arosa record on its own would not be able to succeed or become widely used without collaborations with the WMO Dobson network and NDACC. The recent efforts in the Brewer COST action project are aimed at creating the organized collaborative network of ozone measurements in Europe and homogenization of the near-real time processing of ozone data from multiple Brewer instruments. The effort to retain all individual station data on the same scale was the most important step in creating the homogenized operations for the Dobson network in 1980s.*

Our reply: We tried to cover these points in the last paragraph of Section 8.

We also improved several formulation in the manuscript.

**Referee 2:** I found the paper very interesting, as I'm not a specialist in stratospheric ozone so historical details of how some of the researched developed was new to me. Overall the paper is well written and I only saw a couple of minor grammatical errors. It gives a detailed description of how total column ozone measurements were initiated at Arosa with the aim of studying the impact of mountain air on recovery from tuberculosis and eventually contributed to our understanding of the stratospheric ozone layer and the damage being caused to it by our emissions. The story it tells of pressures from competing scientists and institutes or the difficulties in maintaining funding, illustrate the ways in which science develops in the face of many challenges. It highlights the importance of maintaining high-quality and long-term measurement facilities as the information they give can evolve with time as we become aware of new processes the consequences of human activity. The site is of continued importance for studying the impact of climate change on ozone as well as other factors. My only suggestion would be that the abstract could possibly put a greater emphasis on the value of the historical measurements and continuing them into the future, in the light of climate change. Grammatical notes are in the attached pdf.

Please also note the supplement to this comment: https://www.atmos-chem-phys-discuss.net/acp-2017-1079/acp-2017-1079-RC2supplement.pdf: Could you add a couple of sentences to outline the basic operation of these types of instrument and why the Umkehr was such a significant improvement? Not all readers may be familiar with the optical methods used to measure total column ozone.

Johannes Staehelin, 15.032018

[revised manuscript text omitted]

**1) Arosa Spectrophotometer**
Total Ozone and Umkehr

| Instrument | Characteristic | Ownership | Operation | 1921-30 | 1931-40 | 1941-50 | 1951-60 | 1961-70 | 1971-80 | 1981-90 | 1991-2000 | 2001-10 | 2011-20 |
|---|---|---|---|---|---|---|---|---|---|---|---|---|---|
| Spectrograph[4] | Photographic | LKO Arosa | Campaign[2] | | Occasionally used[3] | | | | | | | | |

[1] Fabry-Buisson type spectrograph build by Schmidt-Haensch (Berlin) from Mar.1926 to Oct. 1928 on a design supervised by Götz, financed by Tourist Office (KVV) Arosa
[2] Instrument oiperated in Spitzbergen 1929 (with D002) [3] Instrument removed of operation after 1954 (exact date not known)

**2) Dobson Spectrophotometers**
a) Total Ozone Measurements (TO)

| | Standard Instr. TO | Manual Operation | Semi-Automated | Fully Automated |

| Instrument | Characteristic | Ownership | Operation | 1921-30 | 1931-40 | 1941-50 | 1951-60 | 1961-70 | 1971-80 | 1981-90 | 1991-2000 | 2001-10 | 2011-20 |
|---|---|---|---|---|---|---|---|---|---|---|---|---|---|
| D002[2] | Photographic | London Met Office | Daily[1] | | Occasionally used | | | | | | | | |
| D007 | Photoelectric | O3 Committee/IMA | Daily[1] | | | | | | | | | | |
| D015 | Photomultiplier | IOC/IMA | Daily[1] | | | | | | | | | | |
| D101[3] | Photomultiplier | ETHZ/MeteoSwiss | Daily[1] | | | | | | | | | | |
| D062 | Photomultiplier | Envir. Canada | Daily[1] | | | | | | | | | | |
| D051[5] | Photomultiplier | IOC/IMA[4] | Daily[1] | | | | | | | | | | |

[1] In favorable weather conditions [2] Féry type spectrograph/name D2 given by Dütsch (not internationally used) /instr. operated in Spitzbergen 1929 (with Arosa Spectrograph)
[3] Since Jan. 2016 operated at PMOD in Davos [4] Intern. O3 Comm./Intern. Met. Association [5] From Jan.1975 to Jun.1985 test operation in fully automated mode b) Umkehr Measurements (UM)

| | Standard Instr. UM | Manual Operation | Semi-Automated | Fully Automated |

| Instrument | Characteristic | Ownership | Operation | 1921-30 | 1931-40 | 1941-50 | 1951-60 | 1961-70 | 1971-80 | 1981-90 | 1991-2000 | 2001-10 | 2011-20 |
|---|---|---|---|---|---|---|---|---|---|---|---|---|---|
| D015 | Photomultiplier | IOC/IMA | Daily[1)(2] | | | | | | | | | | |
| D051 | Photomultiplier | IOC/IMA[4] | Daily[1] | | | | | | | | | | |
| D101[3] | Photomultiplier | MeteoSwiss | Daily[1)(2] | | | | | | | | | | |
| D062 | Photomultiplier | Envir. Canada | Daily[2] | | | | | | | | | | |

[1] In favorable weather conditions [2] Since 1989 only 3 times per month [3] Since Jan. 2016 operated at PMOD in Davos [4] Intern. O3 Comm./Intern. Met. Association

**3) Brewer Spectrophotometers**
Total Ozone, Umkehr and UV spectra[1]

| | | | | | | | | | | | | Fully Automated | |

| Instrument | Type | Ownership | Operation | 1921-30 | 1931-40 | 1941-50 | 1951-60 | 1961-70 | 1971-80 | 1981-90 | 1991-2000 | 2001-10 | 2011-20 |
|---|---|---|---|---|---|---|---|---|---|---|---|---|---|
| Br040 | MarkII[2] | MeteoSwiss | Daily | | | | | | | | | | |
| Br072[3] | MarkII[2] | MeteoSwiss | Daily | | | | | | | | | | |
| Br156 | MarkIII | MeteoSwiss | Daily | | | | | | | | | | |

[1] Up to 2005 Br40 mainly devoted to Totla Ozone and Umkehr, Br72 to Total ozone and Br156 to Total Ozone and UV spectra; in 2005 begin of uniformisation of measuring programmes
[2] MarkII: Single monochromator/ MarkIII: Double monochromator [3] Nov.2011-Mar.2013 and Jun.2014-2017 instrument operated at PMOD in Davos

[revised manuscript text omitted]

- Redundancy allows increasing credibility of measurements, which is particularly important for reliable long-term trend analysis. At Arosa, 3 Dobson and 3 Brewer spectrophotometers were simultaneously operated since 1998, which helped to obtain important scientific results regarding Dobson and Brewer spectrophotometers relevant in the broader context of atmospheric ozone measurements.

- Regular comparison of station instruments with standard spectrophotometers operated under the umbrella of WMO are important for high-quality measurements and comparability of ozone measurements within a particular network.

[revised manuscript text omitted]